# PartSticker: Generating Parts of Objects for Rapid Prototyping

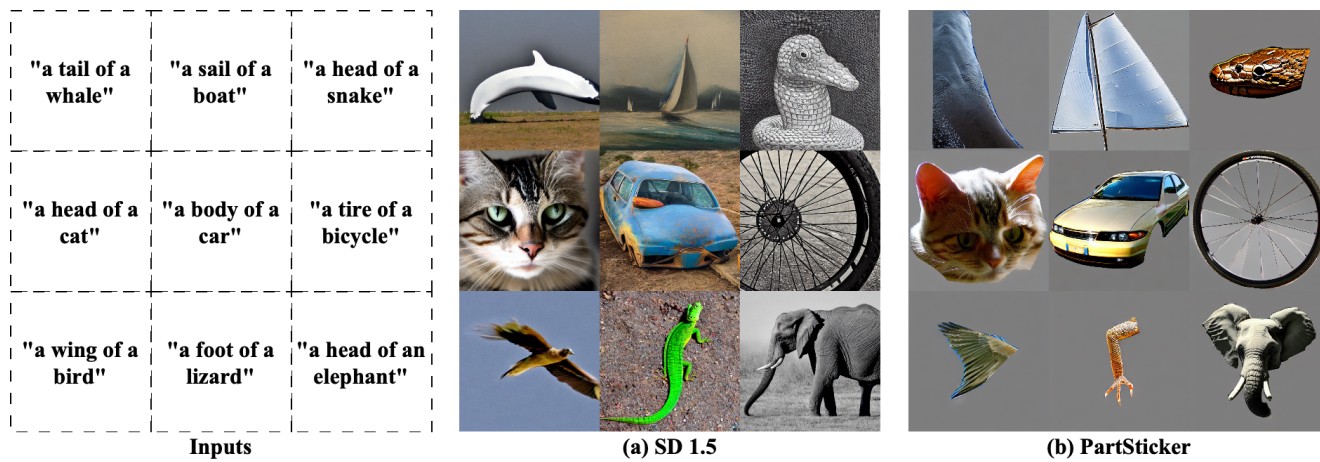

Figure 1. Given text prompts on the left of parts of objects, we show results from (a) a baseline, Stable Diffusion (SD) 1.5, showing that: generated parts can 1) be unrealistic (top), 2) include unwanted information, such as obfuscated content (middle) and 3) show the entire object, rather than just the part of interest (bottom). In contrast, results from (b) our method consistently produce realistic parts on neutral backgrounds, isolated from their parent objects.

## Abstract

*Design prototyping involves creating mockups of products or concepts to gather feedback and iterate on ideas. While prototyping often requires specific parts of objects, such as when constructing a novel creature for a video game, existing text-to-image methods tend to only generate entire objects. To address this, we propose a novel task and method of "part sticker generation", which entails generating an isolated part of an object on a neutral background. Experiments demonstrate our method outperforms state-of-the-art baselines with respect to realism and text alignment, while preserving object-level generation capabilities. We release our code and models at* `https://anonymous.com` *to encourage community-wide progress on this new task.*

## 1. Introduction

Design prototyping involves creating mockups of products or concepts to gather feedback and rapidly iterate on ideas, ultimately refining the final outcome [6, 8, 64]. This low-fidelity approach minimizes the need for polished assets at each step, accelerating creativity and decision-making in fields such as product design, education, and entertainment [3, 12, 54, 55]. Generative AI tools have emerged as a popular tool in rapid prototyping due to their proficiency in generating diverse objects and scenes from linguistic input [4, 7, 23, 37, 48, 58].

Towards increasing the applicability of generative AI for design prototyping, we aim to address three limitations of existing methods. *First*, most text-to-image pipelines cannot directly generate *isolated parts* (e.g., a wing detached from a bird). Yet, such finer-grained, part-level renderings can also be valuable for prototyping, such as for product development. *Second*, existing approaches supporting part-level generation often either compromise on realism or include extra, unwanted regions (e.g., head and shoulders instead of just the head), necessitating extra work before such rendered parts can be incorporated into a design. *Third*, text-to-image pipelines typically produce non-neutral backgrounds, necessitating extra work to isolate rendered parts from the backgrounds. These three limitations are highlighted in **Figure 1(a)**, with results from an existing state-

of-the-art model (Stable Diffusion): the top row shows unrealistic parts; the middle row includes incomplete regions, such as a fraction of the desired tire of the bicycle; and the bottom row displays unwanted extra regions, such as the entire object.

To overcome these limitations, we propose the novel task of *part sticker generation*, which requires a system to create a single, accurately isolated part on a plain background given only a textual prompt. While numerous pay-per-use datasets exist for object-level "stickers" (e.g., `Adobe Stock`, `Noun Project`), free alternatives are limited, and few, if any, offer the ability to generate parts on demand. Our contributions are three-fold. *First*, we introduce **PartSticker**, the *first dedicated method* for isolated part generation that simultaneously provides realism and diversity. *Second*, by generating parts against a neutral background, PartSticker removes the need for manual segmentation to isolate rendered parts, enabling drag-and-drop capabilities that could streamline rapid design prototyping. *Third*, we benchmark our approach against state-of-the-art text-to-image models, identifying common failure modes and discussing avenues for future improvements. As shown in **Figure 1**(b), our method produces realistic, complete, and precisely isolated parts.

Success in our proposed task can yield numerous benefits for the design space. For instance, designers can assemble different generated parts to rapidly prototype novel concepts, similar to how physical LEGO blocks can be combined to create unique structures. Additionally, by simply varying the text prompt, users can quickly produce multiple variations (e.g., shapes, textures, or styles) of a specific part and select the best fit. This low-cost, on-the-fly variation supports rapid A/B testing of design ideas in a visual manner. It also encourages remixing and exploration: for example, a game artist could generate bird, dragon, or airplane wings and attach them to a creature concept, thereby expanding creative possibilities beyond conventional assets.

## 2. Related Work

**Text-to-Image Generation.** Text-to-image generation methods aim to synthesize realistic images from natural language descriptions. There has been rapid progress on this task since the introduction of Generative Adversarial Networks [14] (GANs), an adversarial training framework that pits a generator against a discriminator to produce increasingly plausible images. Building on these foundations, subsequent works incorporated text encoders [27, 44, 47, 57, 66] and attention mechanisms [10, 13, 36, 65] to better align generated content with linguistic input, improving visual fidelity and text relevance. More recently, diffusion [2]-based models have achieved impressive results by iteratively denoising randomly sampled noise [19, 33, 53, 68], leveraging large-scale image-text datasets to produce high-fidelity out-

puts [43, 49]. While these methods excel at generating entire scenes or objects, they offer limited control when users only need to generate specific components, such as parts of an object [51, 60]. In contrast, our work focuses on generating standalone parts directly against a neutral background, eliminating the need for manual intervention while maintaining fidelity to the text prompt.

**Segmentation-Based Generation Models.** Numerous approaches leverage segmentation to guide image synthesis [9, 15, 31, 67]. Early works introduced semantic masks or label maps as constraints to control layouts [34, 62], enabling users to define rough outlines of objects before rendering them [24]. For example, segmentation-aware GANs [25] and conditional architectures can produce images aligned with user-provided masks [41, 52, 68, 69], improving structural accuracy. Although these methods improve realism and controllability, they generally expect explicit segmentation inputs from users (i.e., the models rely on this guidance) and still focus on generating entire objects or complex scenes, rather than just requiring text input. In contrast, our approach capitalizes on part-level annotations during training to directly generate isolated object parts on plain backgrounds when *only given text prompts* (i.e., no segmentation mask inputs).

**Generation of Images with Plain Backgrounds.** Many works aim to produce images with plain or fully controlled backgrounds to support diverse, often creative, tasks like e-commerce photography and compositing. Early techniques often relied on segmentation masks or 'green screen' setups, requiring users to remove unwanted backgrounds by hand before compositing the subject on a plain canvas [1, 39]. More recently, GAN-based methods like SPADE [34] have enabled semantic image synthesis from user-defined segmentation maps, effectively granting fine-grained control over foreground and background regions. Inpainting approaches like InstructPix2Pix [5] use textual prompts to refine or erase backgrounds without altering the main subject. Although these methods provide adjustable backgrounds, many still require substantial user input (e.g., drawing segmentation outlines) or do not offer mechanisms for generating *plain* backgrounds. In contrast, our work automates the plain-background process by directly generating 'stickers', isolated parts placed against a neutral canvas that can be trivially separated by automated methods, thus removing the manual overhead of removing backgrounds by hand.

## 3. Method

We now introduce our **PartSticker** framework, describing its model architecture, data generation pipeline, and training strategies. An overview is shown in **Figure 2**.

### 3.1. Background: Diffusion

Our proposed framework leverages fundamental advancements in diffusion modeling [2, 45]. Diffusion approaches frame image generation as a process of gradually denoising samples, beginning from pure noise and iteratively refining them into coherent images. Following [19], let $\mathbf{x_0} \sim q(\mathbf{x_0})$ be an image sampled from the ground truth distribution (e.g., dataset) $q$. Forward diffusion processes progressively add Gaussian noise over $T$ timesteps, forming a Markov Process with latent variables $\mathbf{x_1}, \ldots, \mathbf{x_T}$. For every time step $t \in \{1, 2, \ldots, T\}$, we sample

$$q(\mathbf{x_t} \mid \mathbf{x_{t-1}}) \sim \mathcal{N}(\sqrt{1 - \beta_t}\mathbf{x_{t-1}}, \beta_t\mathbf{I}), \qquad (1)$$

where $\beta_t$ is the variance scheduler controlling the noise at time step $t$ and $\mathbf{I}$ is the identity matrix with the same dimension as the latents, ensuring that noise is added isotropically (all latent channels are treated equally). This process yields a nearly Gaussian latent vector $\mathbf{x_T}$ after $T$ timesteps, irregardless of $\mathbf{x_0}$.

Diffusion models generally learn the reverse process:

$$p_\theta(\mathbf{x_{t-1}} \mid \mathbf{x_t}), \qquad (2)$$

where $p$ is parametrized by the parameters $\theta$ belonging to a denoising network (e.g., a UNet [46] in our setting). This reverse process iteratively removes noise to reconstruct a de-noised latent vector, $\mathbf{x_0}$. A common realization of this objective is expressed via the simplified denoising loss:

$$\mathcal{L}_{denoise} = \mathbb{E}_{\epsilon \sim \mathcal{N}(0,1), t \sim \mathcal{U}(\{1,\ldots,T\})} \left[ ||\epsilon - \epsilon_\theta(\mathbf{x}_t, t)||_2^2 \right], \qquad (3)$$

where $\epsilon_\theta$ is a time-conditioned network that predicts the Gaussian noise $\epsilon$ at timestep $t$, where $t$ is uniformly sampled from $\{1, \ldots, T\}$. Once the model learns to remove the added noise, the resulting vector $\mathbf{x_0}$ can be decoded back to image space, thereby completing image generation.

### 3.2. PartSticker Framework

The core contribution of our framework is our training data generation pipeline. We design it to support our task of part sticker generation, by generating a single part specified in a text prompt that is 'pasted' on a neutral, gray background. The pipeline is exemplified in **Figure 2**, and consists of two key steps. First, given an image with associated object and part segmentation masks, we localize each desired part and place it in the center of a gray background. Second, for each localized image region, we pair it with a prompt describing the region. All resulting image-prompt pairs are then used to train a text-to-image diffusion model. We hypothesize that such a pipeline would enable a trained model to generate diverse renderings for each part type because existing datasets offer many diverse part segmentations, including of the same types observed across different objects (e.g., an eye can be found in a frog and dog).

**Sticker Construction.** Given a part segmentation dataset with associated images, we create our part stickers by using a templated approach. First, we obtain a binary segmentation for a desired part (e.g., the leg of a dog). We then multiply this binary segmentation with the associated image to obtain a RGB image where pixels outside of the desired region are black and pixels inside are the original content from the image. We then create a blank canvas of square aspect ratio where the background of the canvas is blank and 'paste' the masked content onto the center of the canvas. This design choice stems from the fact that saliency variations are undesired in our setting. Rather, we aim to generate a single, easy-to-extract salient part per image.

**Prompt Generation.** To provide textual conditioning during training, we construct prompts that reflect both the part and the object. Specifically, given an (object, part) pair, such as (dog, leg), we construct our input prompts with the template "a [PART] of a [OBJECT]." We choose this prompt over other choices (e.g., "[OBJECT] [PART].") as language models have shown better hierarchical understanding with the former prompt [30]. However, we hypothesize at test time that users do not need to strictly abide by these prompts due to the model learning the object-part association, and we provide ablation study results that support this hypothesis in **Section 4**.

**Training Approach.** We fine-tune Stable Diffusion 1.5 with our training data in order to extend its powerful base capabilities. We achieve this with Low-Rank Adaptation (LoRA) fine-tuning, by attaching LoRA adapters [20, 40] to the attention layers of the U-Net. LoRA adds low-rank trainable parameters (i.e., matrices of rank $r$) into $Q, K, V$ matrices and output projections (i.e., last linear layer of the attention blocks), which we unfreeze for gradient updates. The rest of the network remains frozen, reducing memory consumption and training time compared to full fine-tuning while preventing catastrophic forgetting. Importantly, this fine-tuning strategy allows us to adapt the network to our part-sticker generation task without discarding the foundational parameters. This approach maintains the model's general "world knowledge" while enabling it to learn domain-specific behaviors pertinent to generating isolated parts on a gray background. During training, we optimize the standard diffusion loss presented in **Equation 3**.

## 4. Experiments

We now assess the performance of our proposed PartSticker framework against modern text-to-image generation baselines, both qualitatively and quantitatively.

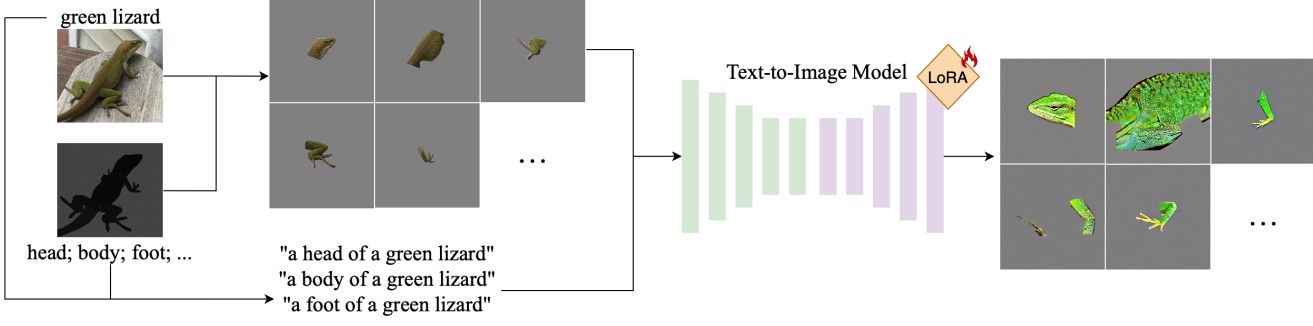

Figure 2. Overview of our proposed **PartSticker** framework. We train a base model on 'part stickers' (i.e., masked out parts of an object pasted on a neutral background) with text prompts describing the region, both of which are derived from existing part segmentation datasets. Text prompts are created by combining the part class labels with their object-level superclasses, leveraging the following template: "a [PART] of a [OBJECT]". We leverage LoRA [20] to achieve parameter-efficient fine-tuning.

| Model | Ours | SD1.5 [45] | SDXL [38] | InstanceDiffusion [61] | GLIGEN [26] |
|---|---|---|---|---|---|
| # Total Parameters | 1.06B | 1.06B | 3.2B | 1.2B | 1.2B |
| Neutral Background Generation | ✓ | ✗ | ✗ | ✗ | ✗ |
| Part-only Generation | ✓ | ✗ | ✗ | ✗ | ✗ |

Table 1. Comparison of our proposed method with other baselines in terms of total parameters, ability to generate regions of interest on a neutral background, and ability to generate solely the region of interest. Out of all methods, PartSticker is the only method capable of achieving the latter two capabilities, and achieves this while also relying on the fewest amount of parameters.

### 4.1. Experimental Design

We evaluate our framework by providing a model with a text prompt describing the part to be generated. We use the prompt introduced in **Section 3**. For each prompt, we consider 100 generated samples from each baseline method.

**Baselines.** We compare our proposed PartSticker framework against numerous popular text-to-image models. A summary of each model's capacity and capabilities is provided in **Table 1**.

First, we leverage Stable Diffusion 1.5 [45] (SD 1.5), which also serves as our base model. SD 1.5 follows a latent diffusion architecture, that compresses images into lower-dimensional latent space via a variational autoencoder [22], then denoises these latent codes with a time-condition U-Net [46], and conditions on text using CLIP [42]. We use the variant of SD 1.5 trained on the large-scale LAION 5B dataset [50], which contains roughly five billion image-text pairs. We include SD 1.5 as a baseline for its proven competence in generating high-fidelity images from text alone while maintaining a relatively lightweight architecture.

We also evaluate Stable Diffusion XL [38] (SDXL), a more parameter-rich extension of SD 1.5 that introduces a refined architecture, multiple text encoders for better language alignment, and a refiner module for enhanced image detail. Trained on an expanded LAION-5B dataset with higher-quality image-text pairs, SDXL offers near state-of-the-art text-to-image performance but demands significantly more computational resources. In our experiments, we compare only to its base model (excluding the refiner) to conserve memory and boost inference speed, while providing a more fair comparison with the other baselines.

Besides text-to-image foundation models, we also compare our method with controllable generative models, which typically offer more precise manipulation of entities in a generated image. We leverage two top-performing models, GLIGEN [26] and InstanceDiffusion [61]. GLIGEN augments a U-Net backbone with learnable 'condition' embeddings and auxiliary modules that take in as input spatial annotations. We use the variant of GLIGEN trained on multiple image-text datasets with spatial annotations. Similarly, InstanceDiffusion integrates a module to handle four types of grounded conditions: bounding boxes, segmentation masks, scribbles, and points, relying on large-scale multimodal annotations. Both models accept text prompts plus one or more conditioning inputs. We use a centered bounding box in all experiments, to yield spatially constrained images.

**Implementation Details.** We implement our PartSticker model using the Diffusers [59] library, alongside PyTorch [35] for model training. To increase training speed,

we leverage mixed-precision. We train our model for 10 epochs using a batch size of 16 text-part sticker pairs and keep the model with the best validation loss. For our LoRA fine-tuning, we use a rank of 16 using a learning rate of $1\mathrm{e}^{-4}$ with the AdamW [28] optimizer with $\beta_1 = 0.9$ and $\beta_2 = 0.999$. We resize all images to a spatial resolution of 512x512. All experiment were conducted on two NVIDIA A100 GPUs. At training time our model takes as input a text prompt and image, while at inference it takes as input a prompt alone.

**Datasets.** To generate our training data, we leverage the high-quality part segmentation dataset, PartImageNet [17], based off of the seminal ImageNet [11] paper. This datasets contains 20k images for training, 1k for validation, and 2k for testing. PartImageNet contains 11 object super-categories which can be further expanded into 158 specific categories (e.g., quadruped to deer) using their Word-Net [29] names. After performing our sticker construction algorithm from **Section 3.2**, we end up with 94k samples for training, 5.6k for validation, and 11k for testing.

Through training and validation, we rely on WordNet-derived object names to construct precise text prompts that preserve rich semantic detail. However, anticipating that real-world users may describe an object region with more general terms (e.g., "car" rather than a specific make like "Subaru"), we have our test-time prompts adopt these more generalized descriptors.

**Evaluation Metrics.** We evaluate our generated part stickers using Fréchet Inception Distance [18] (FID) and Structural Similarity Index Measure [63] (SSIM). FID compares the feature distributions of real and generated images using features extracted from InceptionV3 [56]; values closer to 0 indicate greater similarity to the real distribution, whereas higher values represent signal poorer fidelity. SSIM assesses pixel-wise structural consistency between pairs of images on a scale from -1 to 1, where 1 denotes identical images and -1 denotes perfect anti-correlation (e.g., an inverted version of the image). By examining both FID (global distribution alignment) and SSIM (local structural fidelity), we capture complementary aspects of realism and structural accuracy in our generated outputs.

## 4.2. Results

We now analyze the results of experiments, both quantitatively using traditional image generation evaluation metrics, and qualitatively, using representative generated samples from our baseline methods. We provide quantitative results in **Table 2**, and quantitative results in **Figure 3**.

**Overall Performance.** We report the quantitative performance for our method and all baselines in **Table 2**. For

| Methods | FID↓ | SSIM↑ |
|---|---|---|
| *PartSticker* | **39.52** | **0.74** |
| *Stable Diffusion 1.5* [45] | 81.93 | 0.34 |
| *InstanceDiffusion* [61] | 85.66 | 0.35 |
| *GLIGEN* [26] | 81.62 | 0.35 |
| *SDXL* [38] | 58.37 | 0.48 |

Table 2. We evaluate using FID and SSIM. Our method outperforms prior work across both metrics, indicating its effectiveness on generating isolated part stickers.

both metrics, FID and SSM, PartSticker consistently outperforms the baseline methods by a considerable margin. For example, we achieve a 106% relative percentage point increase in FID over the next best method, GLIGEN, suggesting that our approach more correctly produces images that have a higher degree of realism, as measured by comparing the feature distributions of InceptionV3. Similarly, when observing the next best SSIM scores (i.e., InstanceDiffusion and GLIGEN), we observe a 111% relative percentage point increase in performance. This discrepancy in performance highlights a crucial limitation of existing approaches: they are unable to match patterns in generated images of localized parts, as evidenced by both the low SSIM scores as well as the qualitative examples shown in **Figure 3**.

When observing the quantitative performance of baseline methods, we observe little variance in both SSIM and FID scores, with the exception of InstanceDiffusion, which performs nearly four percentage points worse than the remaining baselines. This poor performance suggests that, while these methods may be able to generate out of distribution objects such as the dinosaur head in **Figure 3**, they are unable to generalize outside of their training data to novel structures such as solely generating a part. The one exception to this trend is SDXL, which attains significantly better FID and SSIM scores than the other baselines, although it still does not surpass our results. Notably, SDXL uses nearly three times as many parameters, granting it substantially more expressive power.

**Analysis with Respect to Specific Parts.** We analyze the effectiveness of our method compared to baseline approaches for generating specific object parts, such as "the tire of a car" or "the wing of a bird," as shown in **Figure 3**. Across all evaluations, our proposed framework demonstrates consistent results, whereas baseline methods yield mixed outcomes. PartSticker consistently generates only the intended part for various object classes. For example, for the prompt "the sail of a boat," our method reliably generates only the sail without including the boat body. Similarly, with the prompt "a foot of a lizard", our method correctly does not include the lizard's body in the generated

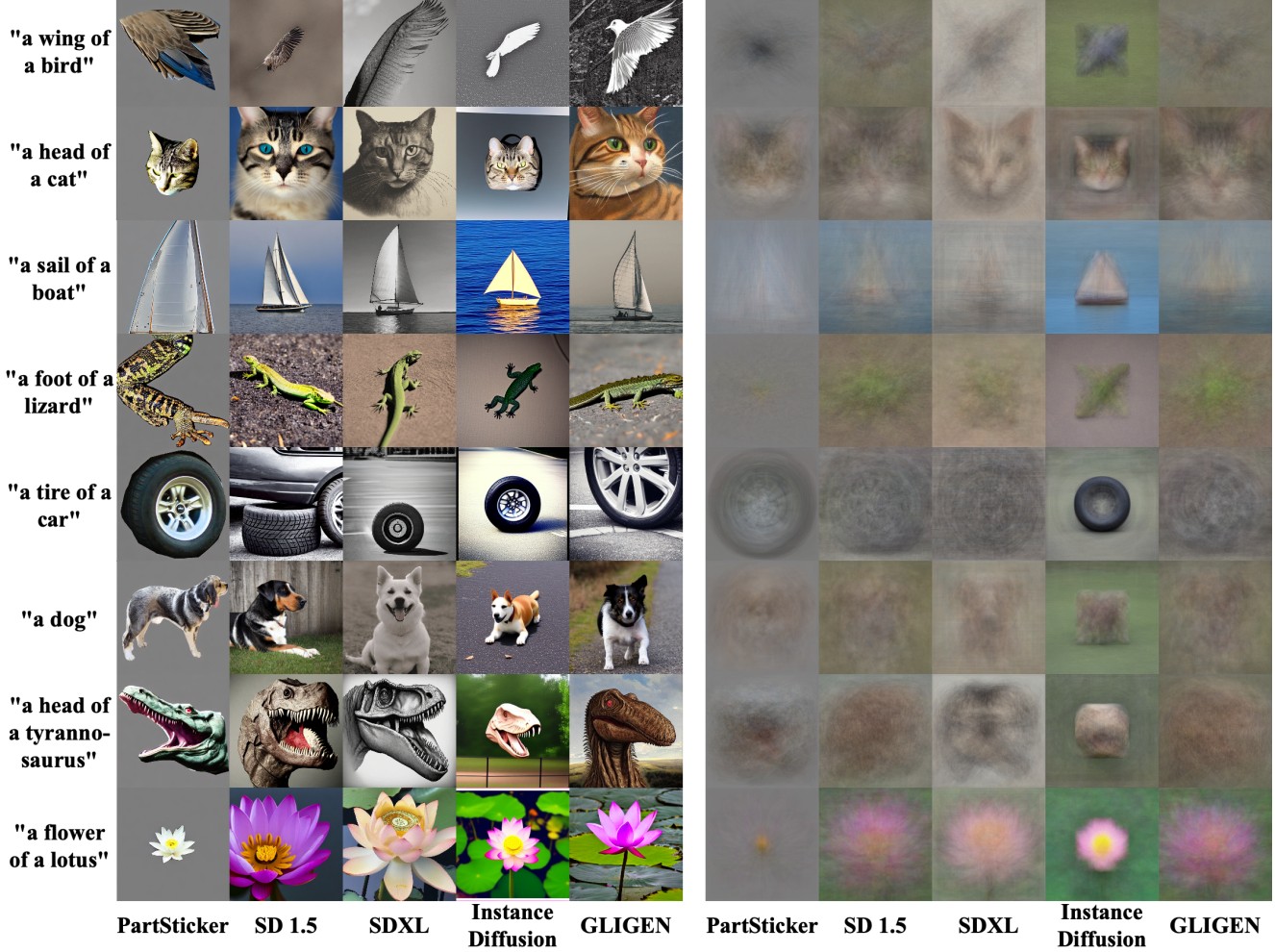

Figure 3. Qualitative results showing examples of generated images given text prompts (left) and the average image of 100 generated samples from a given method (left). Overall, we observe that PartSticker is the only method capable of consistently generating only the requested part on a neutral background with a high degree of realism. The bottom three rows represent out-of-distribution scenarios for PartSticker: generation of an object and two out-of-distribution parts. (SD stands for Stable Diffusion).

content. In contrast, baseline methods frequently generate the entire boat and lizard. Similarly, when prompted to generate "the tire of a car," PartSticker consistently isolates the tire, whereas other methods often include extra portions of a car, such as its fender in one instance. While Stable Diffusion 1.5 and Stable Diffusion XL can occasionally generate only the target part, such as for the prompt "the wing of a bird," such results are inconsistent with many generations containing the full bird. This inconsistency across baseline methods highlights a significant drawback: users cannot reliably predict whether a method will generate solely the desired part. Such unpredictability may necessitate additional effort to isolate the desired parts from the generated images, which we hypothesize is cumbersome for users. In contrast, our method provides consistent and predictable results, reducing the potential need for extra intervention.

**Analysis with Respect to Realism.** Next, we analyze the realism of the generated results with respect to physical accuracy and the diversity between artistic (e.g., sketched or illustrated) and photorealistic content. PartSticker consistently produces physically accurate results. For instance, when prompted with "the foot of a lizard," our method accurately generates the foot without additional limbs or unrealistic artifacts. Baseline methods, such as Stable Diffusion 1.5 and InstanceDiffusion, sometimes generate unrealistic features, including extra limbs or incorrect replacements (e.g., a tail replaced with an additional foot). Likewise, methods like GLIGEN and Stable Diffusion XL occasionally produce unnatural shapes or orientations, such as a distorted car tire or a bird's wing occupying inappropriate body positions, such as the head or the body.

Regarding photorealism, PartSticker consistently gen-

erates photorealistic content, likely benefiting from fine-tuning on real images. In contrast, baseline methods exhibit inconsistent style, alternating between realistic and artistic outcomes, such as sketches. While diversity in style may seem beneficial, we argue that photorealism should be the default output unless explicitly requested by users, as photo-realistic images typically serve broader practical purposes. Other methods either share this inconsistency or produce visibly artificial images.

One limitation of our approach is the harsh edges visible in some stickers, particularly with round parts such as tires, which can appear polygonal. This issue likely arises due to inaccuracies in segmentation masks around region bound-aries during training. Nonetheless, we believe smoothing these edges is less burdensome for users than manually lo-calizing the desired parts, and may be sufficient for most low-fidelity prototypes where demonstrating concepts takes precedence over polished results. Future work could in-volve improving the quality of segmentation datasets or adding refinement modules directly to the generation pro-cess to enhance realism further.

**Analysis with Respect to Saliency.** We analyze the saliency of generated parts by examining the average im-ages on the right of **Figure 3**. To construct the average im-ages, we average 100 generations for each prompt.

Overall, our findings suggest that our method strikes a balance between maintaining a salient focus on the target part and accommodating diverse real-world variations. For example, for prompts such as "a head of a cat" and "a tire of a car," PartSticker consistently produces blurry yet recog-nizable generations corresponding to the specified parts, in-dicating strong saliency. In contrast, for "a wing of a bird," the average image reflects significant variation in perspec-tive and angle, covering a broad range of real-world scenar-ios, an attribute beneficial for design exploration. For the out-of-distribution scenario, "a flower of a lotus," the sta-men is consistently centered, yet the petals remain faint or absent in the averaged result. We hypothesize that, although the model fixes the lotus's position, frequent changes in the petals' sizes, colors, or viewpoints render their contours less defined when averaged.

Most baseline methods have considerably more variation in position than our approach. For example, models such as Stable Diffusion 1.5, SDXL, and GLIGEN produce parts (e.g., "a tire of a car") at diverse positions, as evidenced by most of the average image being the color of a car tire. The one exception is InstanceDiffusion, which is expected be-cause InstanceDiffusion relies on a user-provided bounding box to determine where the generated part should appear, thus consistently centering the target in the image. Interest-ingly, despite GLIGEN receiving similar spatial guidance (i.e., a bounding box), its content exhibits a greater varia-tion in position than InstanceDiffusion. While such spatial variability is potentially desirable for tasks like recognition or segmentation, we hypothesis it is ill-suited for rapid pro-totyping tasks as it may require extra work from a designer to locate the region of interest.

Although one could envision a future hybrid approach that combines bounding-box guidance from InstanceDiffu-sion with our PartSticker framework to offer explicit loca-tion control, it would also require an additional user input (i.e., the bounding box). In comparison, PartSticker cen-ters each generated part by default, balancing user needs for predictable placement and minimal configuration over-head. This design choice obviates the need for bounding-box inputs while enabling part generation capabilities on clean backgrounds.

**Analysis with Respect to Background.** We next ana-lyze the backgrounds associated with the generated content. PartSticker consistently places each generated region of in-terest onto a neutral, gray background across all prompts. Although slight hue variations occur between samples and prompts, PartSticker remains the only method that reliably maintains neutral backgrounds. Baseline methods, in con-trast, generally produce non-neutral backgrounds. Stable Diffusion XL and GLIGEN occasionally produce relatively neutral backgrounds, notably when generating "a head of a cat," but this consistency does not extend to other prompts, creating unpredictable results for end users. InstanceDif-fusion also occasionally generates simpler backgrounds, as seen with in the lizard example, however they often exhibit color gradients or subtle patterns that could complicated au-tomated masking tasks for 'sticker' removal.

**Analysis with Respect to Object Generation.** In the third row from the bottom in **Figure 3**, we show a gener-ated example of an entire object (i.e., a dog) on the left and its 'average image' on the right. Notably, even though our model was not fine-tuned on full-object data, it retains the base model's capacity to generate complete objects. More-over, it exhibits a new capability by placing the dog at the center of a neutral background, suggesting our approach is suitable for rapid prototyping beyond part-level require-ments. The resulting output can be used *as is* with minimal user intervention, unlike the other methods, which depict the dog against a plausible but non-neutral environment.

Observing the average image confirms that most of the generated dog pixels appear near the center, likely a conse-quence of the center-pasting strategy used in our part-sticker training pipeline (**Section 3**). In comparison, the baseline methods (other than InstanceDiffusion, which conditions on a bounding box in the center) exhibit broader variation in the dog's position, leaving the user to locate or isolate the object. As expected, all baselines also generate full ob-

jects effectively, aligning with the tasks they were originally trained for.

**Analysis with Respect to Out-of-distribution Performance.** In the bottom two rows of **Figure** 3, we present out-of-distribution examples for all baseline methods. These prompts are considered out-of-distribution for Part-Sticker because neither the categories (i.e., tyrannosaurus and lotus) nor the text prompts were seen during fine-tuning. They are also out-of-distribution for the baseline methods because their prompts follow the form "a [PART] of [OBJECT]." Notably, when generating "a head of a tyrannosaurus", most methods depict the creature accurately, with only GLIGEN adding extraneous elements such as the neck. When examining background contents, Part-Sticker remains the sole method to produce a fully neutral background. While Stable Diffusion 1.5 and SDXL exhibit gray backdrops, they incorporate vignette effects that cannot be easily removed without access to the RAW image, which is unavailable with generated images since they are not captured from the sensor of a camera. For the lotus example, PartSticker successfully generates the flower alone, whereas most other methods also produce its stem. Despite these categories being unseen during fine-tuning, Part-Sticker maintains strong image quality while preserving a consistent, neutral background.

### 4.3. Model Design Analysis

In this section we ablate key design choices for our Part-Sticker framework to establish their importance.

**Effect of LoRA rank.** Results highlighting the impact of the LoRA rank on fine-tuning our base model are shown in **Table** 3. Intuitively, one might expect higher ranks to yield better performance, since larger LoRA matrices can provide increased capacity. However, our results indicate a more nuanced relationship. When using a rank of 4, the model appears to lack sufficient representational power, leading to the worst performance. Conversely, at rank 32, the model exhibits the second poorest results, suggesting it may be *underfitting* due to excessive capacity relative to our dataset. The best outcomes arise with an intermediate rank of 16, presumably because this configuration strikes an effective balance between representational capacity and demands for our novel task. Notably, all the aforementioned configurations still outperform all baselines by a large margin.

**Effect of amount of training data.** We analyze the impact of training data size in **Table** 4, comparing performance when fine-tuning with 50% of the available data versus the full dataset. As expected, reducing the training data results in a performance drop, consistent with the general trend that foundation models benefit from larger datasets.

| Rank | FID↓ | SSIM↑ |
|------|------|-------|
| 4 | 42.93 | 0.69 |
| 16 | **39.52** | **0.74** |
| 32 | 41.48 | 0.71 |

Table 3. Ablation study on the effect of LoRA rank and data augmentation strategy. A rank of 16 achieves the best performance.

| Amount of Data | FID↓ | SSIM↑ |
|----------------|------|-------|
| 50% | 51.67 | 0.63 |
| 100% | 39.52 | 0.74 |

Table 4. Ablation study on the effect of the amount of available training data for fine-tuning. We observe a decrease in performance when using half the available data, but an increase compared to the baseline methods.

However, a notable observation is that even with only half the data, our model still outperforms all baselines. This finding suggests that while additional data improves performance, modern models require only a fraction of the samples to effectively learn our target task.

## 5. Conclusion

We introduced PartSticker, a specialized generative framework that directly produces isolated object parts with neutral backgrounds from textual prompts. By leveraging LoRA-based fine-tuning on part-segmentation data, Part-Sticker outperforms state-of-the-art methods both quantitatively and qualitatively, which we argue could considerably reduce human effort in rapid prototyping workflows. Future work could explore some method refinements, such as achieving smoother boundaries, as well as investigate compositional strategies for creating entirely new objects from generated parts.

While our method can greatly benefit designers by speeding up iterative workflows, it also carries potential ethical risks, much like other generative AI tools. For instance, such tools can be misused for mass manipulation, disinformation, or producing large volumes of low-quality content [21]. In contrast to image-level generation methods that can instantly generate content for spreading false information, such as Deep Fakes [32], which undermine trust in media [16], our framework requires manual assembly of individual parts into novel designs. This human-in-the-loop process significantly reduces the likelihood of mass-produced misinformation. Additionally, our approach centers on non-human animals and rigid objects [17], limiting its capacity to generate deceptive human imagery while remaining well-suited for rapid prototyping tasks.

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
