# OpenReview forum: "PartSticker: Generating Parts of Objects for Rapid Prototyping"
_thecvf.com/CVPR/2025/Workshop/CVEU — CVPR 2025_

### Official Review · Reviewer_o3cW · 2025-03-16
**Stable diffusion with LoRA trained  for Isolated Object-Part Generation with Neutral Backgrounds**

**Rating:** 5
**Confidence:** 4

**Review:**

### Quality
- The paper is technically sound and demonstrates a strong grasp of diffusion modeling, data preparation, and fine-tuning strategies.
- Quantitative evaluations (FID, SSIM) and qualitative results are solid, showing significant improvements over state-of-the-art.
- Clear ablation studies on LoRA rank and training data size strengthen the empirical grounding.

### Clarity
- The writing is clear and accessible.
- The figures are informative (especially Fig. 1 and Fig. 3), and the methodology is easy to follow.
- Terms like "part sticker" are clearly defined and contextualized.

### Originality
- Introduces a new task: generation of isolated object parts with plain backgrounds.
- Presents a clean, scalable data generation pipeline and shows that minor LoRA tuning suffices for strong performance.
- Offers meaningful analysis of real-world utility for prototyping, separating it from typical text-to-image generation work.

### Significance
- Addresses an unmet need in the design prototyping space — generating precise, editable visual components.
- The method offers practical value in HCI, game design, and rapid concept development.
- Opens the door for future compositional generation pipelines (e.g., building full objects from parts).

---

### Official Review · Reviewer_yCyi · 2025-03-23

**Rating:** 4
**Confidence:** 4

**Review:**

# Idea:
The paper proposes Paart Sticke, to solve the part generation problem needed for idea prototyping. They aim to address three challenges faced by current baselines which include (a) their inability to generate parts rather than whole objects, (b) their  failure to generate part objects in neutral backgrounds, and (c) their inclusion of unwanted areas in generated parts.
# Clarity:
The paper is clear and addresses the part sticker generation problem by proposing a model PartSticker which consists of (a) a stable diffusion 1.5 network and (b) a dataset generation framework for training .
# Novelty and Weaknesses:
1. Seems incremental as it uses previously proposed models and finetuning strategies (LoRA), however uses a new dataset and have a good motivation for the proposed dataset.
2. Seems though the background is not neutral, most images generated by baslines on the part generation task are easy to segment and generally okay? (see figure 3, especially the average images showing that the highlighted areas are generally where the part is, especially for instance diffusion).
3. The variety of different styles of the same part seems not to be explored?
# Quality and Significance:
We believe that despite some drawbacks identified, this work can be useful to the image stylizing and prototyping community.

---

### Official Review · Reviewer_maY3 · 2025-03-25
**Novel part-level image generation task with limitations**

**Rating:** 3
**Confidence:** 3

**Review:**

This paper introduces PartSticker, a technique for generating isolated parts of objects on neutral backgrounds using text prompts. This task is proposed to address limitations in existing text-to-image generation models for design prototyping applications.

The paper identifies a clear gap in current text-to-image generation capabilities and proposes a well-motivated new task of part-level image generation. The implementation builds on top of established diffusion model techniques and is relatively straightforward. The paper also provides detailed quantitative and qualitative comparisons.

While the proposed task is interesting, I do find the scope a bit limiting. For one, would a generative model combined with an object detection and segmentation model achieve a similar or better performance as PartSticker? Considering the motivation of proposing this part sticker task, I feel such implementation may very well work in a prototyping context.
In another example, HoloFusion generates 3D-consistent images from different angles with neutral background. While not specific for generating parts, their results are also relevant to this current paper, and potentially very useful for 3D prototyping.

Overall, while I find the proposed task and motivation interesting and the evaluation solid in general, I do have questions about the usefulness of the proposed task. Therefore, I am on the fence.

---

### Decision · Program_Chairs · 2025-03-25

**Decision:**

Accept

**Comment:**

The paper introduces PartSticker, a novel method addressing a clearly motivated problem—generating isolated object parts on neutral backgrounds for design prototyping. Reviewers acknowledged its originality in defining a new practical task, solid empirical evaluations, and the clarity of the presented method. Though some concerns were raised regarding incremental novelty and limited exploration of alternative approaches, reviewers agreed on the method's practical significance and solid performance gains.

Given the overall positive feedback and clear practical value, the paper is accepted. Authors are encouraged to further clarify the distinct advantages over potential alternative methods, explore style diversity in generated parts, and discuss the implications of combining generative models with object segmentation methods in the camera-ready version.